# Grouping-Based Low-Rank Trajectory Completion and 3D Reconstruction

**Katerina Fragkiadaki**
EECS, University of California,
Berkeley, CA 94720
katef@berkeley.edu

**Marta Salas**
Universidad de Zaragoza,
Zaragoza, Spain
msalasg@unizar.es

**Pablo Arbeláez**
Universidad de los Andes,
Bogotá, Colombia
pa.arbelaez@uniandes.edu.co

**Jitendra Malik**
EECS, University of California,
Berkeley, CA 94720
malik@eecs.berkeley.edu

## Abstract

Extracting 3D shape of deforming objects in monocular videos, a task known as non-rigid structure-from-motion (NRSfM), has so far been studied only on synthetic datasets and controlled environments. Typically, the objects to reconstruct are pre-segmented, they exhibit limited rotations and occlusions, or full-length trajectories are assumed. In order to integrate NRSfM into current video analysis pipelines, one needs to consider as input realistic -thus incomplete- tracking, and perform spatio-temporal grouping to segment the objects from their surroundings. Furthermore, NRSfM needs to be robust to noise in both segmentation and tracking, e.g., drifting, segmentation "leaking", optical flow "bleeding" etc. In this paper, we make a first attempt towards this goal, and propose a method that combines dense optical flow tracking, motion trajectory clustering and NRSfM for 3D reconstruction of objects in videos. For each trajectory cluster, we compute multiple reconstructions by minimizing the reprojection error and the rank of the 3D shape under different rank bounds of the trajectory matrix. We show that dense 3D shape is extracted and trajectories are completed across occlusions and low textured regions, even under mild relative motion between the object and the camera. We achieve competitive results on a public NRSfM benchmark while using fixed parameters across all sequences and handling incomplete trajectories, in contrast to existing approaches. We further test our approach on popular video segmentation datasets. To the best of our knowledge, our method is the first to extract dense object models from realistic videos, such as those found in Youtube or Hollywood movies, without object-specific priors.

## 1 Introduction

Structure-from-motion is the ability to perceive the 3D shape of objects solely from motion cues. It is considered the earliest form of depth perception in primates, and is believed to be used by animals that lack stereopsis, such as insects and fish [1].

In computer vision, non-rigid structure-from-motion (NRSfM) is the extraction of a time-varying 3D point cloud from its 2D point trajectories. The problem is under-constrained since many 3D time-varying shapes and camera poses give rise to the same 2D image projections. To tackle this ambiguity, early work of Bregler *et al.* [2] assumes the per frame 3D shapes lie in a low dimensional subspace. They recover the 3D shape basis and coefficients, along with camera rotations, using a $3K$ factorization of the 2D trajectory matrix, where $K$ the dimension of the shape subspace,

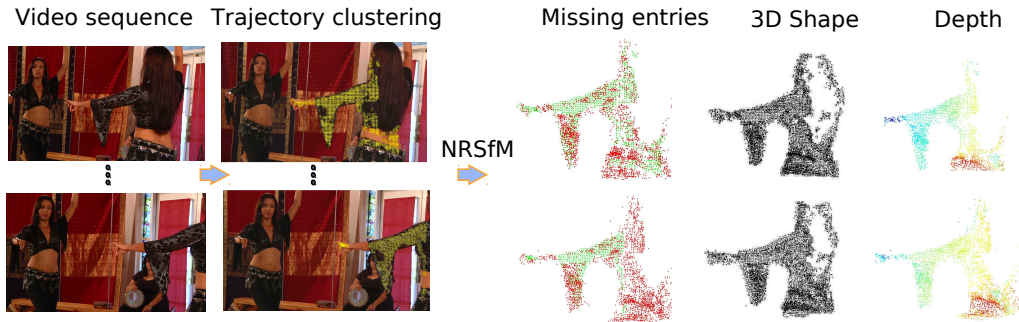

Figure 1: Overview. Given a monocular video, we cluster dense flow trajectories using 2D motion similarities. Each trajectory cluster results in an incomplete trajectory matrix that is the input to our NRSfM algorithm. Present and missing trajectory entries for the chosen frames are shown in green and red respectively. The color of the points in the rightmost column represents depth values (red is close, blue is far). Notice the completion of the occluded trajectories on the belly dancer, that reside beyond the image border.

extending the rank 3 factorization method for rigid SfM of Tomasi and Kanade [3]. Akhter *et al.*[4] observe that the 3D point trajectories admit a similar low-rank decomposition: they can be written as linear combinations over a 3D trajectory basis. This essentially reflects that 3D (and 2D) point trajectories are temporally smooth. Temporal smoothness is directly imposed using differentials over the 3D shape matrix in Dai *et al.* [5]. Further, rather than recovering the shape or trajectory basis and coefficients, the authors propose a direct rank minimization of the 3D shape matrix, and show superior reconstruction results.

Despite such progress, NRSfM has been so far demonstrated only on a limited number of synthetic or lab acquired video sequences. Factors that limit the application of current approaches to real-world scenarios include:

(i) Missing trajectory data. The aforementioned state-of-the-art NRSfM algorithms assume complete trajectories. This is an unrealistic assumption under object rotations, deformations or occlusions. Work of Torresani *et al.* [6] relaxes the full-length trajectory assumption. They impose a Gaussian prior over the 3D shape and use probabilistic PCA within a linear dynamical system for extracting 3D deformation modes and camera poses; however, their method is sensitive to initialization and degrades with the amount of missing data. Gotardo and Martinez [7] combine the shape and trajectory low-rank decompositions and can handle missing data; their method is one of our baselines in Section 3. Park *et al.* [8] use static background structure to estimate camera poses and handle missing data using a linear formulation over a predefined trajectory basis. Simon *at al.* [9] consider a probabilistic formulation of the bilinear basis model of Akhter *et al.* [10] over the non-rigid 3D shape deformations. This results in a matrix normal distribution for the time varying 3D shape with a Kronecker structured covariance matrix over the column and row covariances that describe shape and temporal correlations respectively. Our work makes no assumptions regarding temporal smoothness, in contrast to [8, 7, 9].

(ii) Requirement of accurate video segmentation. The low-rank priors typically used in NRSfM require the object to be segmented from its surroundings. Work of [11] is the only approach that attempts to combine video segmentation and reconstruction, rather than considering pre-segmented objects. The authors projectively reconstruct small trajectory clusters assuming they capture rigidly moving object parts. Reconstruction results are shown in three videos only, making it hard to judge the success of this locally rigid model.

This paper aims at closing the gap between theory and application in object-agnostic NRSfM from realistic monocular videos. We build upon recent advances in tracking, video segmentation and low-rank matrix completion to extract 3D shapes of objects in videos under rigid and non-rigid motion. We assume a scaled orthographic camera model, as standard in the literature [12, 13], and low-rank object-independent shape priors for the moving objects. Our goal is a richer representation of the video segments in terms of rotations and 3D deformations, and temporal completion of their trajectories through occlusion gaps or tracking failures.

An overview of our approach is presented in Figure 1. Given a video sequence, we compute dense point trajectories and cluster them using 2D motion similarities. For each trajectory cluster, we first complete the 2D trajectory matrix using standard low-rank matrix completion. We then recover the camera poses through a rank 3 truncation of the trajectory matrix and Euclidean upgrade. Last, keeping the camera poses fixed, we minimize the reprojection error of the observed trajectory entries along with the nuclear norm of the 3D shape. A byproduct of affine NRSfM is trajectory completion. The recovered 3D time-varying shape is backprojected in the image and the resulting 2D trajectories are completed through deformations, occlusions or other tracking ambiguities, such as lack of texture. In summary, our contributions are:

(i) Joint study of motion segmentation and structure-from-motion. We use as input to reconstruction dense trajectories from optical flow linking [14], as opposed to a) sparse corner trajectories [15], used in previous NRSfM works [4, 5], or b) subspace trajectories of [16, 17], that are full-length but cannot tolerate object occlusions. Reconstruction needs to be robust to segmentation mistakes. Motion trajectory clusters are inevitably polluted with "bleeding" trajectories that, although reside on the background, they anchor on occluding contours. We use morphological operations to discard such trajectories that do not belong to the shape subspace and confuse reconstruction.

(ii) Multiple hypothesis 3D reconstruction through trajectory matrix completion under various rank bounds, for tackling the rank ambiguity.

(iii) We show that, under high trajectory density, rank 3 factorization of the trajectory matrix, as opposed to $3K$, is sufficient to recover the camera rotations in NRSfM. This allows the use of an easy, well-studied Euclidean upgrade for the camera rotations, similar to the one proposed for rigid SfM [3].

We present competitive results of our method on the recently proposed NRSfM benchmark of [17], under a fixed set of parameters and while handling incomplete trajectories, in contrast to existing approaches. Further, we present extensive reconstruction results in videos from two popular video segmentation benchmarks, VSB100 [18] and Moseg [19], that contain videos from Hollywood movies and Youtube. To the best of our knowledge, we are the first to show dense non-rigid reconstructions of objects from real videos, without employing object-specific shape priors [10, 20]. Our code is available at: www.eecs.berkeley.edu/∼katef/nrsfm.

## 2 Low-rank 3D video reconstruction

### 2.1 Video segmentation by multiscale trajectory clustering

Given a video sequence, we want to segment the moving objects in the scene. Brox and Malik [19] propose spectral clustering of dense point trajectories from 2D motion similarities and achieve state-of-the-art performance on video segmentation benchmarks. We extend their method to produce multiscale (rather than single scale) trajectory clustering to deal with segmentation ambiguities caused by scale and motion variations of the objects in the video scene. Specifically, we first compute a spectral embedding from the top eigenvectors of the normalized trajectory affinity matrix. We then obtain discrete trajectory clusterings using the discretization method of [21], while varying the number of eigenvectors to be 10, 20, 30 and 40 in each video sequence.

Ideally, each point trajectory corresponds to a sequence of 2D projections of a 3D physical point. However, each trajectory cluster is spatially surrounded by a thin layer of trajectories that reside outside the true object mask and do not represent projections of 3D physical points. They are the result of optical flow "bleeding " to untextured surroundings [22], and anchor themselves on occluding contours of the object. Although "bleeding" trajectories do not drift across objects, they are a source of noise for reconstruction since they do not belong to the subspace spanned by the true object trajectories. We discard them by computing an open operation (erosion followed by dilation) and an additional erosion of the trajectory cluster mask in each frame.

### 2.2 Non-rigid structure-from-motion

Given a trajectory cluster that captures an object in space and time, let $\mathbf{X}_k^t \in \mathbb{R}^{3 \times 1}$ denote the 3D coordinate $[X \ Y \ Z]^T$ of the $k$th object point at the $t$th frame. We represent 3D object shape with a

matrix $\mathbf{S}$ that contains the time varying coordinates of $K$ object surface points in $F$ frames:

$$\mathbf{S}_{3F \times P} = \begin{bmatrix} \mathbf{S}^1 \\ \vdots \\ \mathbf{S}^F \end{bmatrix} = \begin{bmatrix} \mathbf{X}_1^1 & \mathbf{X}_2^1 & \cdots & \mathbf{X}_P^1 \\ \vdots & & & \vdots \\ \mathbf{X}_1^F & \mathbf{X}_2^F & \cdots & \mathbf{X}_P^F \end{bmatrix}.$$

For the special case of rigid objects, shape coordinates are constant and the shape matrix takes the simplified form: $\mathbf{S}_{3 \times P} = \begin{bmatrix} \mathbf{X}_1 & \mathbf{X}_2 & \cdots & \mathbf{X}_P \end{bmatrix}$.

We adopt a scaled orthographic camera model for reconstruction [3]. Under orthography, the projection rays are perpendicular to the image plane and the projection equation takes the form: $\mathbf{x} = \mathrm{R}\mathbf{X} + \mathbf{t}$, where $\mathbf{x} = [x \ y]^T$ is the vector of 2D pixel coordinates, $\mathrm{R}_{2 \times 3}$ is a scaled truncated rotation matrix and $\mathbf{t}_{2 \times 1}$ is the camera translation. Combining the projection equations for all object points in all fames, we obtain:

$$\begin{bmatrix} \mathbf{x}_1^1 & \mathbf{x}_2^1 & \cdots & \mathbf{x}_P^1 \\ \vdots & \vdots & \vdots & \vdots \\ \mathbf{x}_1^F & \mathbf{x}_2^F & \cdots & \mathbf{x}_P^F \end{bmatrix} = \mathbf{R} \cdot \mathbf{S} + \begin{bmatrix} \mathbf{t}^1 \\ \vdots \\ \mathbf{t}^F \end{bmatrix} \cdot \mathbf{1_P}^T, \tag{1}$$

where the camera pose matrix $\mathbf{R}$ takes the form:

$$\mathbf{R}_{2F \times 3}^{\mathrm{rigid}} = \begin{bmatrix} \mathrm{R}^1 \\ \vdots \\ \mathrm{R}^F \end{bmatrix}, \qquad \mathbf{R}_{2F \times 3F}^{\mathrm{nonrigid}} = \begin{bmatrix} \mathrm{R}^1 & 0 & \cdots & 0 \\ \vdots & \vdots & \cdots & \vdots \\ 0 & 0 & \cdots & \mathrm{R}^F \end{bmatrix}. \tag{2}$$

We subtract the camera translation $\mathbf{t}^t$ from the pixel coordinates $\mathbf{x}^t, t = 1 \cdots F$, fixing the origin of the coordinate system on the objects's center of mass in each frame, and obtain the centered trajectory matrix $\mathbf{W}_{2F \times P}$ for which $\mathbf{W} = \mathbf{R} \cdot \mathbf{S}$.

Let $\tilde{\mathbf{W}}$ denote an *incomplete* trajectory matrix of a cluster obtained from our multiscale trajectory clustering. Let $\mathbf{H} \in \{0, 1\}^{2F \times P}$ denote a binary matrix that indicates presence or absence of entries in $\tilde{\mathbf{W}}$. Given $\tilde{\mathbf{W}}, \mathbf{H}$, we solve for complete trajectories $\mathbf{W}$, shape $\mathbf{S}$ and camera pose $\mathbf{R}$ by minimizing the camera reprojection error and 3D shape rank under various rank bounds for the trajectory matrix. Rather than minimizing the matrix rank which is intractable, we minimize the matrix nuclear norm instead (denoted by $\|\cdot\|_*$), that yields the best convex approximation for the matrix rank over the unit ball of matrices. Let $\odot$ denote Hadamard product and $\|\cdot\|_F$ denote the Frobenius matrix norm. Our cost function reads:

**NRSfM($K$):**

$$\min_{\mathbf{W}, \mathbf{R}, \mathbf{S}} \qquad \|\mathbf{H} \odot (\mathbf{W} - \tilde{\mathbf{W}})\|_F^2 + \|\mathbf{W} - \mathbf{R} \cdot \mathbf{S}\|_F^2 + \mathbf{1}_{K>1} \cdot \mu \|\mathbf{S}^{\mathrm{v}}\|_* \tag{3}$$

$$\text{subject to} \qquad \mathrm{Rank}(\mathbf{W}) \le 3K, \ \exists \alpha_t, \text{s.t. } \mathrm{R}^t(\mathrm{R}^t)^T = \alpha_t I_{2 \times 2}, \ t = 1 \cdots F.$$

We compute multiple reconstructions with $K \in \{1 \cdots 9\}$. $\mathbf{S}^{\mathrm{v}}$ denotes the re-arranged shape matrix where each row contains the vectorized 3D shape in that frame:

$$\mathbf{S}_{F \times 3P}^{\mathrm{v}} = \begin{bmatrix} X_1^1 & Y_1^1 & Z_1^1 & \cdots & X_P^1 & Y_P^1 & Z_P^1 \\ \vdots & \vdots & \vdots & \cdots & \vdots & \vdots & \vdots \\ X_1^F & Y_1^F & Z_1^F & \cdots & X_P^F & Y_P^F & Z_P^F \end{bmatrix} = \begin{bmatrix} P_X & P_Y & P_Z \end{bmatrix} (I_3 \otimes \mathbf{S}), \tag{4}$$

where $P_X, P_Y, P_Z$ are appropriate row selection matrices. Dai *et al.* [5] observe that $\mathbf{S}_{F \times 3P}^{\mathrm{v}}$ has lower rank than the original $\mathbf{S}_{3F \times P}$ since it admits a $K$-rank decomposition, instead of $3K$, assuming per frame 3D shapes span a $K$ dimensional subspace. Though $\mathbf{S}$ facilitates the writing of the projection equations, minimizing the rank of the re-arranged matrix $\mathbf{S}^{\mathrm{v}}$ avoids spurious degrees of freedom. Minimization of the nuclear norm of $\mathbf{S}^{\mathrm{v}}$ is used only in the non-rigid case ($K > 1$). In the rigid case, the shape does not change in time and $\mathbf{S}_{1 \times 3P}^{\mathrm{v}}$ has rank 1 by construction. We approximately solve Eq. 3 in three steps.

**Low-rank trajectory matrix completion** We want to complete the 2D trajectory matrix under a rank bound constraint:

$$\min_{\mathbf{W}} \qquad \|\mathbf{H} \odot (\mathbf{W} - \tilde{\mathbf{W}})\|_F^2$$

$$\text{subject to} \quad \mathrm{Rank}(\mathbf{W}) \le 3K. \tag{5}$$

Due to its intractability, the rank bound constraint is typically imposed by a factorization, $\mathbf{W} = UV^T, U_{2F \times r}, V_{P \times r}$, for our case $r = 3K$. Work of [23] empirically shows that the following regularized problem is less prone to local minima than its non-regularized counterpart ($\lambda = 0$):

$$\min_{\mathbf{W}, \mathbf{U}_{2F \times 3K}, \mathbf{V}_{P \times 3K}} \quad \|\mathbf{H} \odot (\mathbf{W} - \tilde{\mathbf{W}})\|_F^2 + \frac{\lambda}{2}(\|\mathbf{U}\|_F^2 + \|\mathbf{V}\|_F^2)$$
$$\text{subject to} \quad \mathbf{W} = \mathbf{U}\mathbf{V}^T. \tag{6}$$

We solve Eq. 6 using the method of Augmented Lagrange multipliers. We want to explicitly search over different rank bounds for the trajectory matrix $\mathbf{W}$ as we vary $K$. We do not choose to minimize the nuclear norm instead, despite being convex, since different weights for the nuclear term result in matrices of different ranks, thus is harder to control explicitly the rank bound. Prior work [24, 23] shows that the bilinear formulation of Eq. 6, despite being non-convex in comparison to the nuclear regularized objective ($\|\mathbf{H} \odot (\mathbf{W} - \tilde{\mathbf{W}})\|_F^2 + \|\mathbf{W}\|_*$), it returns the same optimum in cases $r >= r*$, where $r*$ denotes the rank obtained by the unconstrained minimization of the nuclear regularized objective. We use the continuation strategy proposed in [23] over $r$ to avoid local minima for $r < r*$: starting from large values of $r$, we iteratively reduce it till the desired rank bound $3K$ is achieved. For details, please see [23, 24].

**Euclidean upgrade** Given a complete trajectory matrix, minimization of the reprojection error term of Eq. 3 under the orthonormality constraints is equivalent to a SfM or NRSfM problem in its standard form, previously studied in the seminal works of [3, 2]:

$$\min_{\mathbf{R}, \mathbf{S}} \quad \|\mathbf{W} - \mathbf{R} \cdot \mathbf{S}\|_F^2$$
$$\text{subject to} \quad \exists \alpha_t, \text{ s.t. } \mathrm{R}^t(\mathrm{R}^t)^T = \alpha_t I_{2 \times 2}, \; t = 1 \cdots F. \tag{7}$$

For rigid objects, Tomasi and Kanade [3] recover the camera pose and shape matrix via SVD of $\mathbf{W}$ truncated to rank 3: $\mathbf{W} = \mathrm{UDV}^T = (\mathrm{UD}^{1/2})(\mathrm{D}^{1/2}\mathrm{V}^T) = \hat{\mathbf{R}} \cdot \hat{\mathbf{S}}$. The factorization is not unique since for any invertible matrix $\mathrm{G}_{3 \times 3}$: $\hat{\mathbf{R}} \cdot \hat{\mathbf{S}} = \hat{\mathbf{R}} \cdot \mathrm{GG}^{-1}\hat{\mathbf{S}}$. We estimate G so that $\hat{\mathbf{R}}\mathrm{G}$ satisfies the orthonormality constraints:

$$\text{orthogonality:} \qquad \hat{\mathbf{R}}_{2t-1}\mathrm{GG}^T\hat{\mathbf{R}}_{2t}^T = 0, \quad t = 1 \cdots F$$
$$\text{same norm:} \qquad \hat{\mathbf{R}}_{2t-1}\mathrm{GG}^T\hat{\mathbf{R}}_{2t-1}^T = \hat{\mathbf{R}}_{2t}\mathrm{GG}^T\hat{\mathbf{R}}_{2t}^T, \quad t = 1 \cdots F. \tag{8}$$

The constraints of Eq. 8 form an overdetermined homogeneous linear system with respect to the elements of the gram matrix $\mathrm{Q} = \mathrm{GG}^T$. We estimate Q using least-squares and factorize it using SVD to obtain G up to an arbitrary scaling and rotation of its row space [25]. Then, the rigid object shape is obtained by $\mathbf{S}_{3 \times P} = \mathrm{G}^{-1}\hat{\mathbf{S}}$.

For non-rigid objects, a similar Euclidean upgrade of the rotation matrices has been attempted using a rank $3K$ (rather than 3) decomposition of $\mathbf{W}$ [26]. In the non-rigid case, the corrective transformation G has size $3K \times 3K$. Each column triplet $3K \times 3$ is recovered independently since it contains the rotation information from all frames. For a long time, an overlooked rank 3 constraint on the Gram matrix $\mathrm{Q}_k = \mathrm{G}_k^T\mathrm{G}_k$ spurred conjectures regarding the ambiguity of shape recovery under non-rigid motion [26]. This lead researchers to introduce additional priors for further constraining the problem, such as temporal smoothness [27]. Finally, the work of [4] showed that orthonormality constraints are sufficient to recover a unique non-rigid 3D shape. Dai et al. [5] proposed a practical algorithm for Euclidean upgrade using rank $3K$ decomposition of $\mathbf{W}$ that minimizes the nuclear norm of $\mathrm{Q}_k$ under the orthonormality constraints.

Surprisingly, we have found that in practice it is not necessary to go beyond rank 3 truncation of $\mathbf{W}$ to obtain the rotation matrices in the case of dense NRSfM. The large majority of trajectories span the rigid component of the object, and their information suffices to compute the objects' rotations. This is not the case for synthetic NRSfM datasets, where the number of tracked points on the articulating links is similar to the points spanning the "torso-like" component, as in the famous "Dance" sequence [12]. In Section 3, we show dense face reconstruction results while varying the truncating rank $\kappa_r$ of $\mathbf{W}$ for the Euclidean upgrade step, and verify that $\kappa_r = 3$ is more stable than $\kappa_r > 3$ for NRSfM of faces.

**Rank regularized least-squares for 3D shape recovery** In the non-rigid case, given the recovered camera poses $\mathbf{R}$, we minimize the reprojection error of the observed trajectory entries and 3D shape

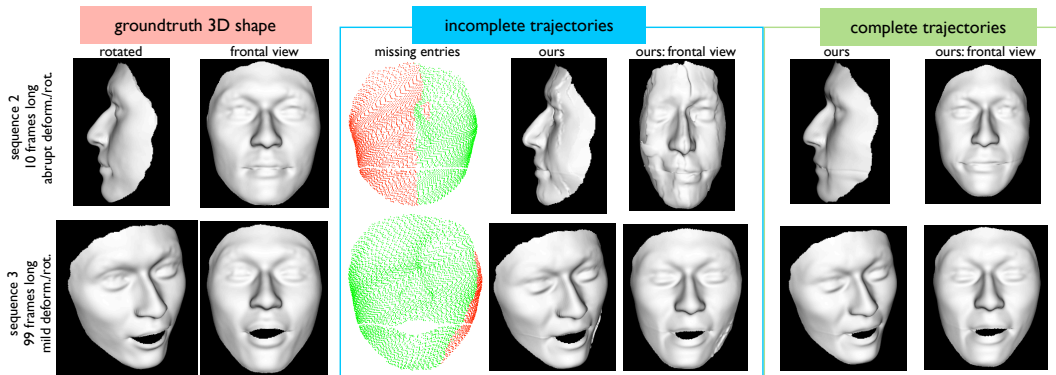

Figure 2: Qualitative results in the synthetic benchmark of [17]. High quality reconstructions are obtained with oracle (full-length) trajectories for both abrupt and smooth motion. For incomplete trajectories, in the 3rd column we show in red the missing and in green the present trajectory entries. The reconstruction result for the 2nd video sequence that has 30% missing data, though worse, is still recognizable.

nuclear norm:

$$\min_{\mathbf{S}}. \quad \frac{1}{2}\|\mathbf{H} \odot (\tilde{\mathbf{W}} - \mathbf{R} \cdot \mathbf{S})\|_F^2 + \mu\|\mathbf{S}^{\text{v}}\|_*$$
$$\text{subject to} \quad \mathbf{S}^{\text{v}} = [P_X \quad P_Y \quad P_Z] (I_3 \otimes \mathbf{S}). \tag{9}$$

Notice that we consider only the observed entries in $\tilde{\mathbf{W}}$ to constrain the 3D shape estimation; however, information from the complete $\mathbf{W}$ has been used for extracting the rotation matrices $\mathbf{R}$. We solve the convex, non-smooth problem in Eq. 9 using the nuclear minimization algorithm proposed in [28]. It generalizes the accelerated proximal gradient method of [29] from $l_1$ regularized least-squares on vectors to nuclear norm regularized least-squares on matrices. It has a better iteration complexity than the Fixed Point Continuation (FPC) method of [30] and the Singular Value Thresholding (SVT) method [31].

Given camera pose $\mathbf{R}$ and shape $\mathbf{S}$, we backproject to obtain complete centered trajectory matrix $\mathbf{W} = \mathbf{R} \cdot \mathbf{S}$. Though we can in principle iterate over the extraction of camera pose and 3D shape, we observed benefits from such iteration only in the rigid case. This observation agrees with the results of Marques and Costeira [32] for rigid SfM from incomplete trajectories.

## 3   Experiments

The only available dense NRSfM benchmark has been recently introduced in Garg *et al.* [17]. They propose a dense NRSfM method that minimizes a robust discontinuity term over the recovered 3D depth along with 3D shape rank. However, their method assumes as input full-length trajectories obtained via the subspace flow tracking method of [16]. Unfortunately, the tracker of [16] can tolerate only very mild out-of-plane rotations or occlusions, which is a serious limitation for tracking in real videos. Our method does not impose the full-length trajectory requirement. Also, we show that the robust discontinuity term in [17] may not be necessary for high quality reconstructions.

The benchmark contains four synthetic video sequences that depict a deforming face, and three real sequences that depict a deforming back, face and heart, respectively. Only the synthetic sequences have ground-truth 3D shapes available, since it is considerably more difficult to obtain ground-truth for NRSfM in non-synthetic environments. Dense full-length ground-truth 2D trajectories are provided for all sequences. For evaluation, we use the code supplied with the benchmark, that performs a pre-alignment step at each frame between $\mathbf{S}^t$ and $\mathbf{S}_{GT}^t$ using Procrustes analysis. Reconstruction performance is measured by mean RMS error across all frames, where the per frame RMS error of a shape $\mathbf{S}^t$ with respect to ground-truth shape $\mathbf{S}_{GT}^t$ is defined as: $\frac{\|\mathbf{S}^t - \mathbf{S}_{GT}^t\|_F}{\|\mathbf{S}_{GT}^t\|_F}$.

Figure 2 presents our qualitative results and Table 1 compares our performance against previous state-of-the-art NRSfM methods: Trajectory Basis (TB) [12], Metric Projections (MP) [33], Variational Reconstruction (VR) [17] and CSF [7]. For CSF, we were not able to complete the experiment for sequences 3 and 4 due to the non-scalable nature of the algorithm. Next to the error of each

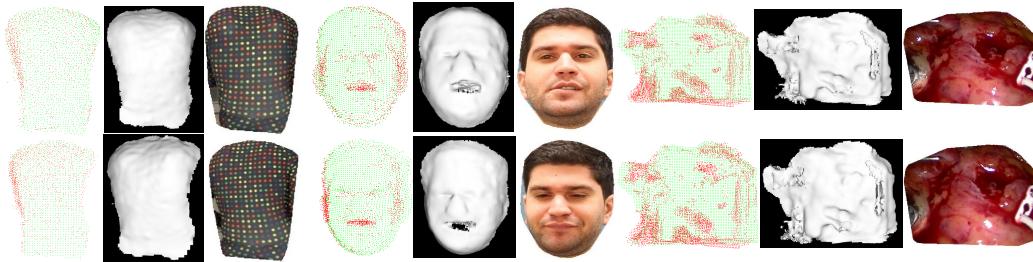

Figure 3: Reconstruction results in the "Back", "Face" and "Heart" sequences of [17]. We show present and missing trajectory entries, per frame depth maps and retextured depth maps.

method we show in parentheses the rank used, that is, the rank that gave the best error. Our method uses exactly the same parameters and $K = 9$ for all four sequences. Baseline VR [17] adapts the weight for the nuclear norm of $\mathbf{S}$ for each sequence. This shows robustness of our method under varying object deformations. $\kappa_r$ is the truncated rank of $\mathbf{W}$ used for the Euclidean upgrade step. When $\kappa_r > 3$, we use the Euclidean upgrade proposed in [5]. $\kappa_r = 3$ gives the most stable face reconstruction results.

Next, to imitate a more realistic setup, we introduce missing entries to the ground-truth 2D tracks by "hiding" trajectory entries that are occluded due to face rotations. The occluded points are shown in red in Figure 2 3rd column. From the "incomplete trajectories" section of Table 1, we see that the error increase for our method is small in comparison to the full-length trajectory case.

In the real "Back", "Face" and "Heart" sequences of the benchmark, the objects are pre-segmented. We keep all trajectories that are at least five frames long. This results in $29.29\%, 30.54\%$ and $52.71\%$ missing data in the corresponding trajectory matrices $\tilde{\mathbf{W}}$. We used $K = 8$ for all sequences. We show qualitative results in Figure 3. The present and missing entries are shown in green and red, respectively. The missing points occupy either occluded regions, or regions with ambiguous correspondence, e.g., under specularities in the Heart sequence.

Next, we test our method on reconstructing objects from videos of two popular video segmentation datasets: VSB100 [18], that contains videos uploaded to Youtube, and Moseg [19], that contains videos from Hollywood movies. Each video is between 19 and 121 frames long. For all videos we use $K \in \{1 \cdots 5\}$. We keep all trajectories longer than five frames. This results in missing data varying from $20\%$ to $70\%$ across videos, with an average of $45\%$ missing trajectory entries. We visualize reconstructions for the best trajectory clusters (the ones closest to the ground-truth segmentations supplied with the datasets) in Figure 4.

**Discussion** Our 3D reconstruction results in real videos show that, under high trajectory density, small object rotations suffice to create the depth perception. We also observe the tracking quality to be crucial for reconstruction. Optical flow deteriorates as the spatial resolution decreases, and thus high video resolution is currently important for our method. The most important failure cases for our

| | ground-truth full trajectories | | | | | | incomplete trajectories | |
|---|---|---|---|---|---|---|---|---|
| | TB [12] | MP [33] | VR [17] | ours $\kappa_r = 3$ | ours $\kappa_r = 6$ | ours $\kappa_r = 9$ | ours $\kappa_r = 3$ | CSF |
| Seq.1 (10) | 18.38 (2) | 19.44 (3) | 4.01 (9) | 5.16 | 6.69 | 21.02 | 4.92 (8.93% occl) | 15.6 |
| Seq.2 (10) | 7.47 (2) | 4.87 (3) | 3.45 (9) | 3.71 | 5.20 | 25.6 | 9.44 (31.60% occl) | 36.8 |
| Seq.3 (99) | 4.50 (4) | 5.13 (6) | 2.60 (9) | 2.81 | 2.88 | 3.00 | 3.40 (14.07% occl) | —— |
| Seq.4 (99) | 6.61 (4) | 5.81 (4) | 2.81 (9) | 3.19 | 3.08 | 3.54 | 5.53 ( 13.63% occl) | —— |

Table 1: Reconstruction results on the NRSfM benchmark of [17]. We show mean RMS error per cent (%). Numbers for TB, MP and VR baselines are from [17]. In the first column, we show in parentheses the number of frames. $\kappa_r$ is the rank of $\mathbf{W}$ used for the Euclidean upgrade. The last two columns shows the performance of our algorithm and CSF baseline when occluded points in the ground-truth tracks are hidden.

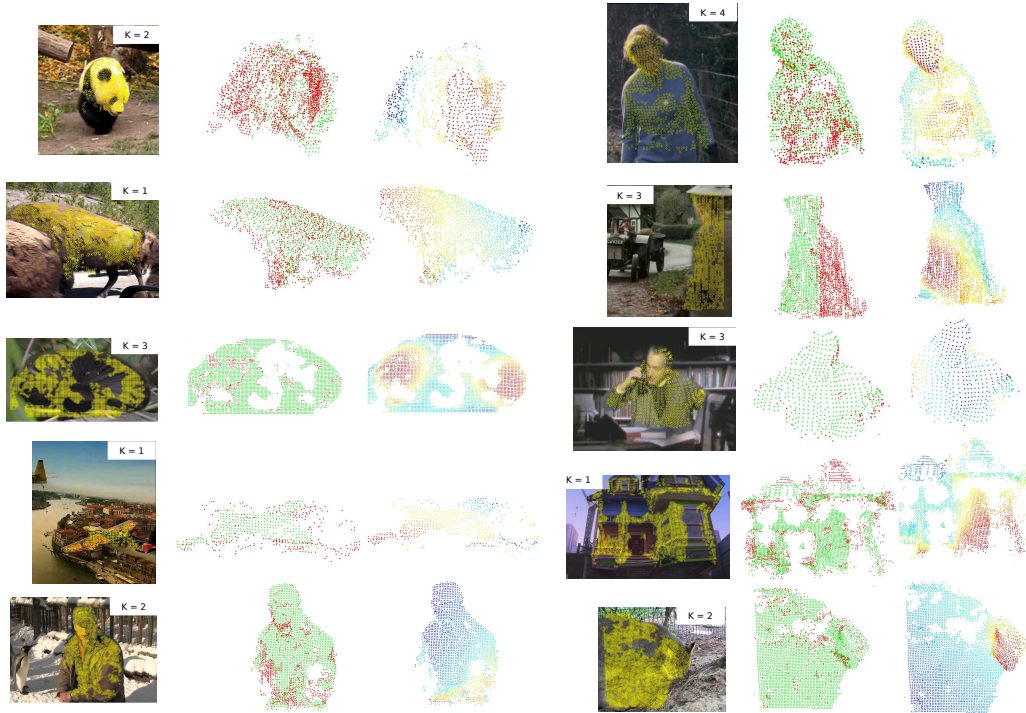

Figure 4: Reconstruction results on the VSB and Moseg video segmentation datasets. For each example we show a) the trajectory cluster, b) the present and missing entries, and c) the depths of the visible (as estimated from ray casting) points, where red and blue denote close and far respectively.

method are highly articulated objects, which violates the low-rank assumptions. 3D reconstruction of articulated bodies is the focus of our current work.

## 4    Conclusion

We have presented a practical method for extracting dense 3D object models from monocular uncalibrated video without object-specific priors. Our method considers as input trajectory motion clusters obtained from automatic video segmentation that contain large amounts of missing data due to object occlusions and rotations. We have applied our NRSfM method on synthetic dense reconstruction benchmarks and on numerous videos from Youtube and Hollywood movies. We have shown that a richer object representation is achievable from video under mild conditions of camera motion and object deformation: small object rotations are sufficient to recover 3D shape. "We see because we move, we move because we see", said Gibson in his "Perception of the Visual World" [34]. We believe this paper has made a step towards encompassing 3D perception from motion into general video analysis.

**Acknowledgments**

The authors would like to thank Philipos Modrohai for useful discussions. M.S. acknowledges funding from Dirección General de Investigación of Spain under project DPI2012-32168 and the Ministerio de Educación (scholarship FPU-AP2010-2906).

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
