[Reviews · NeurIPS 2014]

Submitted by Assigned_Reviewer_13

The paper presents a method for simultaneous dense 3D reconstruction and segmentation of deformable 3D structures in video. This problem has been studied in the past but the present paper presents results on more realistic sequences (from Hollywood movies and Youtube) than earlier work. Furthermore, the method performs at par with the state of the art on standard benchmark datasets with ground truth.

Quality
The paper is technically sound and refers to the relevant previous work. The experiments are reproducable from the description in the paper. However, the experiments are lacking in the sense that there is no baseline or goodness measure for the reconstruction from the realistic sequences. How would the previous methods have performed on the same experiments. Maybe a qualitative analysis of this would be enough.

Clarity
The paper is very clear, well written and well organized.

Originality
The paper is incremental in the sense that it addresses a previously well studied problem. The contribution is more in terms of engineering and method.

Significance
The problem of structure from motion is important in many fields in Computer Vision and Robotics. A method for robust reconstruction of non-rigid structures from video is thus an important contribution. However, due to the lack of comparative results on the realistic sequences, it is hard to tell how far the paper advances the search for such a method.
Summary: The paper, as well as the work described, is of high quality. However, it is hard to judge the significance without a baseline on the realistic sequences.

Submitted by Assigned_Reviewer_27

The paper introduces a new method for non-rigid structure from motion (NRSfM). Existing work on NRSfM typically make assumptions about their inputs, and usually require complete point trajectories, perfect segmentations, or strong object-specific shape priors. The proposed method overcomes these limitations. The method achieves competitive results on an existing synthetic NRSfM metric, and is applied to real-world videos.

Quality: Overall the quality is acceptable, but there are some concerns about the experiments that should be addressed:

- The only quantitative comparison of the proposed method with existing methods is on the synthetic dataset introduced in [11], where the proposed method is provided with full-length trajectories. In this experiment the method from [11] outperforms the proposed method on all video sequences; this is somewhat concerning.

- Even so, the proposed method is not directly comparable to the methods of [6], [28], and [11] since the proposed method can be used even in the presence of incomplete trajectories and imperfect segmentations. The authors claim that these benefits make their method useful for real-world video sequences, but there is no quantitative evaluation of these claims. The authors provide reconstruction results of their method on example videos from the VSB100 and Moseg datasets, and the reader is expected to qualitatively assess these results. A quantitative metric for evaluating these results would be much more convincing.

Clarity: Overall the paper does a good job explaining the method and comparing it to existing work. The description of the method is mostly clear up to line 206; after that the explanation becomes more dense and harder to follow. Some specific comments:

- Why are Equations 5 and 6 solved via successive subproblems? Is this standard in the literature? Would it be possible to optimize jointly over all variables using some convex relaxation of Equations 5 and 6?

- Typo on line 232: W = UDV^T = UDV^T = …

- The Euclidean upgrade step is confusing. This could be a result of my unfamiliarity with the literature, but this section is difficult for me to follow. Why is the rank of W truncated? Why is orthogonality of \hat{R}_{2t-1}G and \hat{R}_{2t}G desirable?

- You claim that your work shows that a rank 3 decomposition of W is better than the standard rank 3K decomposition used in prior work. Do you have any explanation for why this might be the case? Do higher values of K_r lead to overfitting? If so, could this effect be mitigated somehow?

- What is the intuition behind minimizing \|S^V\|_\ast? My (possibly incorrect) interpretation is that when S^V is low rank, the object can be described by a small number of planar regions which do not deform too much between frames. Is really a type of shape or deformation prior? If so how does it differ from the shape priors used in other NRSfM methods?

Originality: I am not very familiar with existing SfM methods or literature, so it is hard for me to accurately assess the originality of the proposed method.

Significance: Nonrigid structure from motion from real-world monocular video sequences is an an important problem. The method proposed by this paper claims to make sizeable advances toward solving this problem. If these claims are valid then this paper is quite significant.

Summary: The proposed method claims to have significant benefits over existing methods, but these claims are not sufficiently backed up by quantitative experimental results.

Submitted by Assigned_Reviewer_30

This paper is about non rigid structure from motion technique (nrsfm). The peculiarity of this paper is that it makes possible to apply nrsfm to standard video sequences and it does not require pre-segmented objects or artificial scenes. This is a very good point of the paper.
In this work they propose a method that makes use of optical flow tracking for motion trajectory clustering and nrsfm for 3D reconstruction of objects in videos.

The paper is well motivated, technically sound, and it presents interesting theoretical novelties. Experiments though are only suboptimal. This work is the first to try nrsfm 'in the wild'.
It is very interesting and a good practical point, that the authors try to remove the trajectories that bleed from the object to the background. This is beneficial to the reconstruction quality.
Interestingly, the authors propose a dense reconstruction of the object and a rank 3 factorization of the trajectory matrix.

Even though the experiments show nrsfm in youtube clips or other movies, it is still unclear how this method works in case of multiple objects moving in the video stream. All the examples are concerned with only one object in the screen. How are these selected? Does it come 'naturally' from the algorithm. If yes, how? Clarifying is very important to show the potential impact of this paper.
Evaluation are not clear, experiments are shown for standard nrsfm datasets and also in youtube clips. It is still unclear why this method is always less performant than VR. I understand that no parameters have to be changed: what happens with optimal params? How much is better than state of the art?
Another unclear point is why the reconstruction in seq1 is better with incomplete traj than with the full ones. Please explain.
These points are important to assess the quality and impact of this work.
Summary: The peculiarity of this paper is that it makes possible to apply nrsfm to standard video sequences and it does not require pre-segmented objects. The paper is well motivated, technically sound, and it presents interesting theoretical novelties. Experiments though are only suboptimal and need clarification to understand the quality of the paper wrt state of the art.
Author Feedback
Author rebuttal: The reviewers have found our approach “well motivated, technically sound, and presenting interesting theoretical novelties”, “well executed”, the paper “very well written”, the experiments “reproducible from the description in the paper”.

We would like first to clarify our quantitative evaluation on the reconstruction benchmark of [11]. The reviewers notice that the previous state of the art (VR[11]) outperforms our approach in the case of full length trajectories (Table 1). The superior performance of VR[11] is due to parameter finetuning in each video sequence. Although the comparison is unfair for our automated approach, we obtain results comparable to [11]. For more details, please see Figure 2d of [11], which depicts the reconstruction error of their method while varying the weight of the spectral shape norm. For example, in that plot, the error for Seq. 3 ranges between 2.6 and 3.3, a range which contains our error of 2.81.

Furthermore, full length trajectories that extend through object occlusions can only be obtained with the help of an oracle that sees through solid objects. Assuming trajectories that track accurately occluded object parts is unrealistic (as acknowledged by the authors of [11]) and beyond the reach of real-world trackers. In this light, the results obtained with full trajectories are not that relevant, since they cannot be achieved without oracle information. Our method obtains comparable performance using only visible trajectories on this benchmark (in green in Fig 2), as shown in the last column of Table 1 and in the supplementary videos.

To quantify our improvement over previous work in 3D reconstruction under occlusions, we have tested on the same benchmark the approach of Gotardo and Martinez TPAMI2011 (entitled “Computing Smooth Time-Trajectories for Camera and Deformable Shape in Structure from Motion with Occlusion”) using their publicly available code. It is a NRSFM method that has shown results on reconstructing lab-acquired 2D trajectories with missing entries. However, in their paper, the occlusions are almost always simulated by randomly discarding trajectory points. This is unrealistic since occlusions cause structured rather than random missing entries. After exhaustive search over the 4 parameters of their algorithm, we obtained 15.6% and 36.8% error for Seq. 1 and 2 using the same incomplete input, much higher than our corresponding errors of 4.92% and 9.44% respectively. We were not able to complete the experiment for Seq 3 and 4 after several hours, due to the non scalable nature of the algorithm. Our method is scalable and has only one parameter (K). We will include the comparison with Gotardo-Martinez in the paper.

Potential impact:
Our approach is the first to tackle NRSFM ‘in the wild’, away from controlled laboratory conditions and oracle assumptions. Given a monocular video, it segments the moving objects and provides access to their dense 3D shapes and per frame camera rotations, and completes correspondences through self-occlusions. These capabilities are critical in any video analysis pipeline and we expect our work to have a large practical impact on the computer vision community.

Reviewers would like to see quantitative evaluation on realistic video sequences. As in all previous NRSFM work, we are restricted by the great difficulty in acquiring groundtruth data for this task. The only annotations available are either from motion capture[6] or from synthetic data[11]. Motion capture provides only sparse 3D trajectory groundtruth which is not suitable for our method, and we already show results on the synthetic benchmark of [11]. Although improving NRSFM benchmarks is not the focus of the present paper, we acknowledge the necessity of more realistic NRSFM benchmarks and they will be part of our future work.

R30: Our method handles multiple objects present in the video. Our spectral clustering partitions the point trajectories into clusters that span the different objects. Each trajectory cluster is reconstructed separately for the low rank assumptions to hold. We show only the foreground segmentations and reconstructions in each video.

R27: Why rank 3 decomposition of W is better than 3K for computing rotations?
This result was for us an empirical observation and a rather unexpected one! The explanation we have in the paper is that the majority of trajectories lie in the rigid torso-like component of the object, which is sufficient to provide the object rotations. So, indeed, the rank 3 truncation acts as a form of regularization, since articulated object parts cannot provide good information of the object’s viewpoint, i.e. camera pose, but rather confuse that estimation.

R27: Why orthogonality?
Orthogonality is desirable since we are trying to recover rotation matrices whose rows should be orthogonal, truncation helps to eliminate noise, as in [2].

R27: Shape prior?
We assume the 3D shapes in time lie in a low dimensional subspace, similar to [3,5,11,23]. S^v allows to go to a lower rank (K rather than 3K) and has been previously used in [5].

R27:Joint optimization?
We could not find a joint way of optimizing our objective. It is indeed typical in NRSFM literature to first compute camera poses and then, given those, to estimate the 3D shape.

R30 asks why in Seq1 the error using incomplete trajectories is slightly lower than that of full trajectories. We currently do not have yet a good explanation for that, we did repeat the experiment multiple times and got the same result. The difference between the two reconstructions is practically unnoticeable, as shown in the supplementary video.

R27: After Ln206 the presentation is dense.
We will include a supplementary file to detail the Euclidean upgrade step of our approach, which is similar to the one used in [2,21]. That would make the paper easier to read.
Thank you very much for your input.